# Electrical Stimulation of Adipose-Derived Stem Cells in 3D Nanofibrillar Cellulose Increases Their Osteogenic Potential

**DOI:** 10.3390/biom10121696

**Published:** 2020-12-18

**Authors:** Mesude Bicer, Jonathan Sheard, Donata Iandolo, Samuel Y. Boateng, Graeme S. Cottrell, Darius Widera

**Affiliations:** 1Stem Cell Biology and Regenerative Medicine Group, School of Pharmacy, University of Reading, Whiteknights, Reading RG6 6UB, UK; M.Bicer@pgr.reading.ac.uk (M.B.); jonathan@sheardbiotech.com (J.S.); 2Sheard BioTech Ltd., 1st Floor Sheraton House, Lower Road, Chorleywood WD3 5LH, UK; 3Mines Saint-Etienne, University of Lyon, Université Jean Monnet, UMR INSERM U1059, 158 Cours Fauriel, 42023 Saint-Etienne, France; donata.iandolo@emse.fr; 4School of Biological Sciences, University of Reading, Whiteknights, Reading RG6 6UB, UK; s.boateng@reading.ac.uk; 5Cellular and Molecular Neuroscience, School of Pharmacy, University of Reading, Reading RG6 6UB, UK; g.s.cottrell@reading.ac.uk

**Keywords:** electrical stimulation, osteogenic differentiation, adipose-derived stem cells, 3D cell culture, nanofibrillar cellulose

## Abstract

Due to the ageing population, there is a steadily increasing incidence of osteoporosis and osteoporotic fractures. As conventional pharmacological therapy options for osteoporosis are often associated with severe side effects, bone grafts are still considered the clinical gold standard. However, the availability of viable, autologous bone grafts is limited making alternative cell-based strategies a promising therapeutic alternative. Adipose-derived stem cells (ASCs) are a readily available population of mesenchymal stem/stromal cells (MSCs) that can be isolated within minimally invasive surgery. This ease of availability and their ability to undergo osteogenic differentiation makes ASCs promising candidates for cell-based therapies for bone fractures. Recent studies have suggested that both exposure to electrical fields and cultivation in 3D can positively affect osteogenic potential of MSCs. To elucidate the osteoinductive potential of a combination of these biophysical cues on ASCs, cells were embedded within anionic nanofibrillar cellulose (aNFC) hydrogels and exposed to electrical stimulation (ES) for up to 21 days. ES was applied to ASCs in 2D and 3D at a voltage of 0.1 V/cm with a duration of 0.04 ms, and a frequency of 10 Hz for 30 min per day. Exposure of ASCs to ES in 3D resulted in high alkaline phosphatase (ALP) activity and in an increased mineralisation evidenced by Alizarin Red S staining. Moreover, ES in 3D aNFC led to an increased expression of the osteogenic markers osteopontin and osteocalcin and a rearrangement and alignment of the actin cytoskeleton. Taken together, our data suggest that a combination of ES with 3D cell culture can increase the osteogenic potential of ASCs. Thus, exposure of ASCs to these biophysical cues might improve the clinical outcomes of regenerative therapies in treatment of osteoporotic fractures.

## 1. Introduction

In healthy individuals, bone tissue undergoes continuous remodelling which involves bone resorption by osteoclasts and osteoblast-mediated deposition of new bone material. In ageing individuals, this balanced process is biased towards bone resorption resulting in reduction of bone density and an increased prevalence of osteoporosis and risk of bone fractures [1]. Indeed, it has been reported that between 30 to 50% of people over 50 develop osteoporotic fractures [2]. Although several pharmacological treatment options for osteoporosis including bisphosphonates, calcitonin and oestrogen-like drugs have been developed, they are often poorly tolerated and are frequently associated with side effects including but not limited to inflammation, severe musculoskeletal pain, and osteonecrosis [3]. Therefore, surgical procedures including autologous bone grafts and allograft implantations are widely used to treat osteoporotic fractures [4]. However, even though bone grafting is currently considered the gold standard in the treatment of complicated osteoporotic fractures, the use of this technique is limited by high incidence of morbidity and lack of donor material [5,6,7]. To overcome these limitations, several cell-based therapies have been suggested including the use of stem cells and progenitor cells either alone or in a combination with biomaterials [8].

Due to their intrinsic ability to differentiate into the osteogenic lineage as well their immunomodulatory and anti-inflammatory potential, along with the potential to be used in an autologous setup, bone marrow-derived mesenchymal stromal/stem cells (BM-MSCs) represent the most promising cell type to achieve bone regeneration in an osteoporotic environment [9,10]. However, there is increasing evidence that in osteoporosis, the number of BM-MSCs, their differentiation capacity, and anti-inflammatory potential are markedly reduced [11,12,13,14]. As this can be attributed to a shift from osteoblastogenesis to predominant adipogenesis in the bone marrow [1], MSCs from different sources including ASCs could represent a viable cellular alternative.

Osteogenic differentiation of ASCs and MSCs can be induced by a combination of soluble factors such as dexamethasone, ascorbic acid, and β-glycerophosphate and further increased by a wide range of biophysical factors including nano-, micro-, and macrotopography [15,16], substrate stiffness [17], fluid shear stress [18], nanovibration-induced displacement [19,20], and ES [21,22]. The effects of ES on cells have been recently reviewed and they appear to include a profound impact on cellular proliferation and osteogenic differentiation [23]. Sundelacruz et al. showed that the membrane potential of MSCs is tightly linked to the differentiation ability and to their preference towards an adipogenic or an osteogenic lineage [24,25]. Short-term ES has been shown to increase the osteogenic potential beyond the treatment period [26]. Moreover, ES has been suggested as a potential approach to treat osteoporotic bone fractures [26]. In 2007, Sun et al. exposed human BM-MSCs to ES with 0.1–1 V/cm and observed that both conditions result in a decrease of calcium oscillation and an increase of osteogenic differentiation [27]. Importantly, no significant differences in osteogenic induction potential were observed between 0.1V and 1V. Although higher voltages are still used in the field [21,28,29], their long-term application can result in electrochemical reaction in the medium and increased cell death rates [30].

Most studies assessing MSC and ASC differentiation are traditionally conducted on flat two-dimensional (2D) substrates that do not closely mirror the physiological and pathophysiological bone niche. In this context, it has been reported that 2D culture can result in a loss of multipotency, premature cellular senescence, and accumulation of chromosomal aberrations within the MSC genome [31,32,33]. In contrast, 3D cell culture not only mimics the 3D structure of the bone tissue but can also positively influence osteogenic differentiation of MSCs whilst simultaneously reducing their cellular senescence [34]. MSCs of different tissue origin have been successfully cultivated in a multitude of different solid 3D scaffolds and hydrogels including calcium phosphate ceramic [35], apatite-wollastonite [36], poly(ε-caprolactone) [37], alginate [38] collagen [39], Matrigel [40], bacteria-derived cellulose [41], and methylcellulose [42].

In our previous studies, we showed that hydrogels based on plant-derived nanofibrillar cellulose (NFC) [43] as well as aNFC hydrogels represent suitable scaffolds for the expansion and osteogenic differentiation of ASCs in 3D [44]. Briefly, we demonstrated that NFC does not negatively affect proliferation of ASCs [43], whilst aNFC significantly increases their viability compared to 2D controls [44].

In the present study, we hypothesized that combining 3D cultivation of ASCs in aNFC hydrogels with ES can positively influence their osteogenic differentiation potential. We have shown that ES of ASCs can increase their osteogenic differentiation in 2D whilst decreasing the levels of adipogenic differentiation. This work also identified that ES of ASCs embedded in 3D aNFC hydrogels increases their osteogenic differentiation compared to unstimulated 3D controls.

## 2. Materials and Methods

### 2.1. aNFC

aNFC (GrowDexT^®^) was kindly provided by UPM Biochemicals, Helsinki, Finland. The handling and preparation of the 0.2% aNFC hydrogel was performed as previously described [44].

### 2.2. Human ASCs

Human ASCs from 3 nondiabetic adult donor lipoaspirates characterised at passage 1 were obtained from Lonza (Slough, UK). All ASCs have been characterised immunocytochemically and by tri-lineage differentiation assay as detailed by the manufacturer and as recommended by The International Society for Cellular Therapy [45].

### 2.3. Cultivation of ASCs as 2D Monolayer

ASCs from three individual donors were cultivated in DMEM high glucose, 2 mM L-glutamine, 100 U penicillin with 100 μg/mL streptomycin (all from Sigma-Aldrich, Gillingham, UK), 20% *v/v* heat-inactivated FBS, (Sigma-Aldrich, lot: 8204188981), and 5 ng/mL basic fibroblast growth factor (Peprotech, London, UK) [standard medium]. Cells were cultured in a humidified incubator (BINDER APT.lineTM C150) at 37 °C and 10% CO_2_. Medium was changed every 2–3 days. All cells were used between passages 7 and 11. For biological replicates, ASCs within a range of 3 passages were used.

### 2.4. Osteogenic and Adipogenic Differentiation in 2D

ASCs were plated in standard medium into tissue culture treated 6-well plates at a density of density of 3.3 × 10^3^/cm^2^. After 72 h, medium was replaced by StemPro^®^ Osteocyte basal medium supplemented with StemPro^®^ osteogenesis supplement [osteogenic medium] or StemPro^®^ adipocyte differentiation basal medium supplemented with StemPro^®^ adipogenesis supplement [adipogenic medium] according to manufacturer’s instructions (all Life Technologies, Thermo Fisher Scientific, Renfrew, UK). The experimental design is shown in Figure 1. Cells were cultivated for up to 21 days in a humidified incubator at 37 °C and 5% CO_2_. Medium was changed every three days.

### 2.5. ALP Activity in 2D

ASCs were subjected to osteogenic differentiation or cultivated in standard medium for 7 days. Activity of ALP was assessed using the Alkaline Phosphatase Diethanolamine Detection Kit (Sigma-Aldrich) including p-nitrophenyl phosphate (p-NPP) as a substrate according to manufacturer’s instructions. Absorbance was measured at a wavelength of 405 nm using a SpectraMax iD3 plate reader (Molecular Devices).

### 2.6. Alizarin Red S Staining in 2D

ASCs differentiated for 21 days were fixed for 15 min using 4% paraformaldehyde (PFA) followed by 3 wash steps using PBS with 5 min per wash step. Calcium deposition was visualised by staining the cells with 1% Alizarin Red S in double deionized water (ddH_2_O, Sigma-Aldrich) at pH 4.3 for 5 min at room temperature followed by imaging using a Nikon A1R inverted confocal microscope (Nikon, Surbiton, UK). Alizarin Red S-based quantification of calcium deposition was performed as described elsewhere [46].

### 2.7. Oil Red O Staining in 2D

ASCs were subjected to adipogenic differentiation as described above and processed for Oil Red O staining as described in [47]. For spectrometric quantification of Oil Red O, cells were fixed with 4% PFA for 30 min followed by elution using 100% 2-propanol. Absorbance was measured at a wavelength of 540 nm using a SpectraMax iD3 plate reader (Molecular Devices). For microscopic analysis of the lipid droplets, cells were fixed and stained with Oil Red O as described above and images were taken using an EVOS XL microscope (Thermo Fisher Scientific). Droplet size was analysed using Fiji (a packaged version of ImageJ [48]).

### 2.8. ES in 2D

For ES in 2D, ASCs were plated into 6-well tissue culture (TC) treated non-pyrogenic polystyrene plate (Corning) at a density of 3.3 × 10^3^/cm^2^ and placed into an IonOptix C-pace EP system (IonOptix LLC, Westwood, MA, USA). ES was conducted as described earlier [49] with some modifications. Briefly, ES cells were paced with a frequency of 10 Hz, pulse duration of 0.04 ms and a voltage of 0.1 V/cm for 30 min per day for up to 21 days.

### 2.9. Cultivation of ASCs in 3D aNFC Hydrogels

After detachment with 0.05% trypsin/EDTA (Sigma-Aldrich), ASC suspensions were mixed with stock aNFC at 1% *w/v* to obtain a desired hydrogel concentration of 0.2% *w/v* aNFC with cell densities of 5 × 10^5^ cells/mL. ASCs in 0.2% aNFC were cultivated in TC cell culture inserts (24-well format, 3.0 μm pores, 83.3932.300, Sarstedt, 100 µL/insert).

### 2.10. ES of ASCs in 3D

ASCs in 0.2% aNFC were cultivated in perforated TC inserts and 3 inserts containing 100 µL of the hydrogel were placed into each well of a 6-well tissue culture treated non-pyrogenic polystyrene plate. Stimulation was performed for up to 21 days using IonOptix C-pace EP system as described above.

### 2.11. Osteogenic Differentiation, ALP Activity Assay, and Alizarin Red S Staining in 3D

For osteogenic differentiation in 3D, ASCs (5 × 10^5^ cells/mL) in standard medium were embedded in aNFC and 100 µL of the suspension was placed into TC inserts as described above. Medium was changed after 24 h to osteogenic medium. For the control cells in 3D, medium was changed to fresh standard medium. Control and differentiation media were replaced every 3 days and maintained at 37 °C and 5% CO_2_ for up to 21 days. Following the differentiation period, cells were washed with PBS and fixed with 4% PFA for 30 min. For the assessment of ALP activity, cells were cultivated for 7 days, and washed with PBS. Detection of ALP activity was performed using the Alkaline Phosphatase Diethanolamine Detection Kit (Sigma-Aldrich) including p-nitrophenyl phosphate substrate (p-NPP). Briefly, 100 μL of lysis buffer (1% Triton X-100 in PBS) was added to the cell samples, incubated for 30 min, and the lysate was transferred to a 96-well plate and equilibrated at 37 °C for 5 min. Absorbance was measured at a wavelength of 405 nm using a SpectraMax iD3 plate reader (Molecular Devices). For Alizarin Red staining, Alizarin Red S staining solution (Sigma-Aldrich) at pH 4.3 was added to cells and incubated for 45 min at room temperature (RT) at dark. The staining solution was removed, and unbound dye was washed off by 5 washing steps with ddH_2_O. Images were taken using a Nikon A1R inverted confocal microscope (Nikon). Quantification of Alizarin Red S signal was performed as described earlier [50].

### 2.12. Viability Assays

XTT assays and calcein/ethidium homodimer-1 live/dead assays in 2D and 3D were performed as described elsewhere [44].

### 2.13. Immunocytochemistry

ASCs were cultured in appropriate medium in 0.2% aNFC in TC inserts for up to 21 days followed by fixation using 4% PFA for 20 min at RT. Unspecific antibody binding was blocked by 0.02% PBS-Tween with 5% horse serum for 30 min. Primary antibody against osteocalcin (OCN, 1:100, mouse monoclonal, clone G-5, Santa Cruz Biotechnology Inc., Heidelberg, Germany), and osteopontin (OPN, 1:100, mouse monoclonal, clone AKm2A1, Santa Cruz Biotechnology Inc.) were added in 0.02% PBS-Tween with 5% horse serum and incubated overnight at 4 °C in agitation. Cells were then washed with PBS overnight at 4 °C in agitation followed by incubation with donkey α-mouse Alexa Fluor 555 secondary antibody (Thermo Fisher Scientific, 1:300) in 0.02% PBS-Tween with 5% goat serum overnight at 4 °C in agitation. Cells were then washed twice with PBS and incubated in PBS overnight at 4 °C in agitation. Cells were counterstained with DAPI (1:2000, Sigma-Aldrich) in PBS overnight at 4 °C in agitation and washed twice with PBS. Following the final wash, PBS was replaced with 0.02% PBS-sodium azide and the samples were imaged using a Nikon A1R inverted confocal microscope (Nikon).

### 2.14. F-actin Staining

Staining of F-actin was conducted as described in [44], with minor modifications. Briefly, phalloidin-Atto 555 (Sigma-Aldrich) was used to label the actin filaments of fixed ASCs cultured in 0.2% aNFC within TC inserts. Following fixation, cells were washed with PBS for 5 min and permeabilised using PBS/0.1% Triton X-100 (100 μL) per well for 30 min at RT. Staining solution (100 μL) was added to the cells and incubated overnight at 4 °C. Following overnight incubation, staining solutions were removed, and the cells were washed once with PBS. Following the final wash, PBS was replaced with 0.02% PBS-sodium azide and the samples were imaged using a Nikon A1R inverted confocal microscope (Nikon).

### 2.15. Statistical Analysis

Statistical analyses were performed using GraphPad Prism software (GraphPad version 8.4.3, San Diego, CA, USA). Data were compared using either Student’s *t*-test (two-tailed, confidence interval 95%) or one-way analysis of variance (ANOVA) with Bonferroni correction (confidence interval 95%), where appropriate. At least 3 independent experiments were performed in triplicate. *p* < 0.05 was considered statistically significant.

## 3. Results

### 3.1. ES Increases Osteogenic Differentiation Potential of ASCs under 2D Cell Culture Conditions

To verify that ASCs display increased osteogenic differentiation without affecting their viability after exposure to ES in 2D, cells in standard medium and osteogenic differentiation medium were stimulated with 0.1 V/cm for up to 21 days followed by assessment of viability at day 7 (d7), day 14 (d14), and day 21 (d21) using an XTT assay. As an early marker of osteogenic differentiation, ALP activity was measured at d7. Alizarin Red S staining was used to assess the levels of mineralisation, whereas immunocytochemistry and subsequent fluorescence microscopy were used for detecting the expression levels of the osteogenic markers OCN and OPN at d21.

Measuring the cellular metabolic activity as an indicator of cell viability, XTT assay revealed no significant differences in ASC viability at d7 independent of the cultivation medium and the presence or absence of ES (Figure 2A). A small but significant decrease in viability was observed in cells cultivated under differentiation conditions with ES compared to cells in standard medium exposed to ES at d14 (Figure 2B). However, no significant differences in cell viability between the experimental groups were observed at d21 (Figure 2C).

Additional live/dead assays at d21 revealed no differences in viability between cells in standard medium and osteogenic medium without ES (Figure 2D). A significantly higher viability was observed in cells exposed to ES and osteogenic medium compared to cells treated with ES in standard medium (Figure 2D).

Assessment of the ALP activity at d7 revealed no significantly higher ALP activity in ASCs cultivated under osteogenic conditions, whereas cells subjected to electrical stimulation in osteogenic differentiation medium showed significantly higher ALP activity than both cells subjected to ES in standard medium and cells in osteogenic medium without ES (Figure 2E). To assess the levels of mineralisation, cells were lysed at d21 and the absorbance of Alizarin Red S was measured by photometry. ASCs subjected to ES showed the overall highest level of mineralisation compared to all other groups (Figure 2F). In addition, significant differences were observed between cells subjected to osteogenic differentiation with and without ES. Immunocytochemical analysis of the expression levels of OCN at d7, d14, and d21 revealed a significantly higher expression levels in cells under osteogenic conditions with and without ES (Figure 2G–I). At d14 and d21, ES resulted in increased OCN expression in cells in osteogenic medium compared to cells cultivated under osteogenic conditions without ES (Figure 2G–I).

Increased levels of OPN in cells in osteogenic medium compared to cells in standard medium were detected at d7 and d14 (Figure 2J,K). In contrast, expression levels of OPN were not found to be significantly different between all conditions although a trend towards higher expression was observed in cells in osteogenic differentiation medium at d21 (Figure 2L).

### 3.2. ES Decreases Adipogenic Differentiation of ASCs under 2D Cell Culture Conditions

ES has been reported to interfere with adipogenic differentiation of human ASCs [43]. To validate these findings in our experimental system, ASCs were subjected to adipogenic differentiation with and without ES for 21 days.

At d21, adipogenic differentiation was assessed by determining the absorbance of lipid droplets stained with Oil Red O and by microscopy-based analysis of lipid droplet size. ES of ASCs under adipogenic differentiation conditions resulted in slight and non-significant decrease of Oil Red O absorbance compared to cells cultivated in adipogenic differentiation medium without ES (Figure 3A,B). However, the average size of lipid droplets decreased in cells subjected to ES in the presence of adipogenic medium as compared to both cells under adipogenic conditions without ES and control cells (Figure 3C).

### 3.3. Long-Term Cultivation of ASCs in Anionic Nanofibrillar Cellulose Does Not Affect Their Viability under Standard and Osteogenic Differentiation Conditions

To determine if long-term cultivation of ASCs under standard conditions in 3D aNFC hydrogels affects their viability, cells were cultivated for up to 21 days followed by assessment of cellular viability using XTT assays. Analysis of XTT data revealed that 3D cultivation in aNFC did not change viability of ASCs compared to 2D conditions when cultivated for 7, 14, and 21 days (Appendix A).

### 3.4. ES of ADSCs Embedded in 3D aNFC Hydrogel Only Moderately Decreases Their Viability under Standard and Osteogenic Conditions

In order to assess the effects of ES on ASC viability in 3D under standard and osteogenic differentiation conditions, cells were cultivated in the respective medium for 7, 14, and 21 days followed by assessment of viability using XTT assays. On d21, additional live/dead staining with subsequent confocal laser scanning microscopy was performed. While no differences in cell viability were detected at d7 and d14, a small but significant decrease was observed in cells cultivated under osteogenic conditions with ES compared to cells in 3D with ES on d21 (Figure 4A–C).

However, there was no significant difference in viability between cells in osteogenic medium with and without ES (Figure 4C). To validate the surrogate viability data obtained in the XTT assay, a live/dead assay was also performed (Figure 4D). Confocal laser scanning microscopy and image analysis revealed higher relative viability in 3D cells in osteogenic differentiation medium compared to control cells in aNFC without ES. Similarly, a higher relative viability was observed in cells subjected to ES under osteogenic differentiation conditions as compared to cells in 3D stimulated with ES in standard medium (Figure 4E). A significant decrease of viability was detected in cells in standard medium with ES compared to cells in standard medium without exposure to ES. Similarly, we detected a lower viability in cells with ES in osteogenic medium compared to differentiated cells without ES (Figure 4E).

### 3.5. ASCs Exposed to ES in 3D Show Increased ALP Activity and Higher Levels of Calcium Deposition during Osteogenic Differentiation

In order to determine early osteogenic differentiation, ALP activity was assessed at d7. We observed that ALP activity was upregulated with and without ES compared to their respective controls (Figure 5A). As a marker of late osteogenesis, levels of calcium deposition were measured using Alizarin Red S with subsequent confocal laser scanning microscopy (Figure 5B,C).

Image-based analysis of Alizarin Red S autofluorescence revealed a higher level of mineralisation in cells under osteogenic conditions compared to the controls with and without ES (Figure 5B,C). Notably, cells in osteogenic medium subjected to ES exhibited a significantly higher signal than cells subjected to differentiation without ES (Figure 5B).

### 3.6. ES under Osteogenic Conditions in 3D Increases Expression of OCN But Does Not Affect OPN Level in ASCs

To study the effects of ES on the expression of the osteogenic markers OCN and OPN, cells were differentiated in 3D aNFC hydrogels for up to 21 days followed by immunocytochemical staining, confocal laser scanning microscopy, and subsequent image analysis.

Consistent with the results obtained in 2D, ES resulted in an increase of OCN expression in cells under osteogenic conditions compared to their respective controls at d14 and d21 (Figure 6B–D). Moreover, at d21, cells under osteogenic conditions with ES exhibited significantly higher levels of OCN compared to cells in osteogenic medium without stimulation (Figure 6D).

Exposure of ASCs to ES resulted in an increased expression of OPN in cells in osteogenic medium compared to cells in osteogenic medium without ES at d14 (Figure 7C). In contrast, no significant differences in OPN expression between all conditions were found at d21 (Figure 7D).

### 3.7. ASCs Exposed to ES Display Highly Arranged Actin Cytoskeleton and Formation of Cell-Free Pores within the 3D aNFC Hydrogel

Increased actin polymerization with perinuclear actin bundles framing the nucleus and alignment of actin fibres in cell periphery are well characterised hallmarks of osteogenesis [51], reviewed in [52]. To visualise the actin cytoskeleton, ASCs under standard and osteogenic conditions with and without ES were fixed and stained with phalloidin at d21 (Figure 8).

Image analysis showed highly aligned actin cytoskeleton in cells subjected to osteogenic differentiation with and without ES. Overall, as shown by the more intense staining, a denser actin cytoskeleton was detected in cells with ES in both standard and osteogenic conditions.

## 4. Discussion

In this study, we investigated the impact of ES on ASCs embedded within 3D aNFC hydrogels. ES of ASCs as well as human and rat BM-MSCs in 2D has been reported to facilitate their osteogenic differentiation [21,27,53]. To validate these findings, ASCs were cultivated as conventional 2D monolayers and exposed to an electric field of 0.1 V/cm. We observed that ES does not affect the viability of the cells under standard cultivation conditions up to 21 days in culture, whereas a small but significant reduction of viability was observed under osteogenic conditions at d14. ASCs in 2D exposed to ES showed increased ALP activity, calcium deposition, and an upregulation of OCN expression at d21. In contrast, upregulation of OPN was observed at d7 and persisted until d14. This is in general accordance with the finding that during in vitro differentiation, OPN is upregulated as early as 7 days and subsequently gradually downregulated, whereas OCN is a late marker of osteogenic differentiation which is still expressed at high levels at d21 [54]. In addition to its influence on osteogenesis, ES has been shown to interfere with adipogenic differentiation of ASCs [55]. Similarly, in our hands, ES of ASCs in 2D significantly reduced the size of lipid droplets in cells subjected to adipogenic differentiation. Interestingly, in the osteoporotic bone marrow, a shift from osteoblastogenesis to adipogenesis negatively affects matrix formation and mineralization via lipotoxic effects [1]. Furthermore, osteoporosis has a negative impact on the number and osteogenic differentiation potential of MSCs [12] in favour of adipogenesis [56,57]. Based on our 2D data, ES could be used to reverse this phenomenon by increasing osteogenesis and decreasing adipogenic differentiation of MSCs.

Recently, we have shown that short-term cultivation of ASCs in plant-derived 3D aNFC hydrogels increases their viability [44]. In contrast to other hydrogels, plant-derived aNFC does not require cross-linking, is optically transparent and has high batch-to-batch consistency. Moreover, NFC-based hydrogels have been shown to support growth of human MSCs and have been successfully applied in clinical settings [58].

In order to determine if long-term cultivation of ASCs in 0.2% aNFC affects their viability, cells were embedded into the hydrogels, cultivated in standard medium and viability assessed using XTT assays at d7, d14, and d21. Data analysis revealed that 3D cultivation of ASCs in aNFC does not affect their viability at all investigated time points. Combining 3D cell culture in aNFC with osteogenic differentiation conditions and ES did not result in significant differences in cell viability detected by XTT at d7 and d14. However, a small but significant decrease of viability was observed in cells cultivated under osteogenic conditions with ES compared to cells in 3D without ES on d21. As XTT assays measure mitochondrial enzyme activity as a surrogate for cellular viability, this can lead to false positive and false negative results. Therefore, ethidium homodimer-1/calcein-based live/dead staining was used to validate the results of the XTT assays. Similar to the XTT results, a slight reduction of cellular viability was observed in cells subjected to ES in both standard and osteogenic differentiation medium compared to the respective non-stimulated counterparts. These results are in agreement with previous reports indicating that the exposure of 2D-cultured MSCs to ES for up to 14 days results in a slight reduction in cell viability [21].

Concomitant use of osteogenic differentiation medium and ES has been shown to increase ALP activity and calcium deposition in human BM-MSCs in 2D [27]. In our study, we assessed ALP activity, calcium mineralisation, and expression levels of the osteogenic markers OPN and OCN in ASCs subjected to osteogenic differentiation and ES in 3D aNFC hydrogels. In contrast to the 2D study by Sun and colleagues [27], exposure of ASCs to ES for 7 days did not significantly increase the ALP activity in cells cultivated in the presence of osteoinductive factors comparted to cells in osteogenic medium without ES. This discrepancy could be explained by cell type specific differences (ASCs vs. BM-MSCs) or as earlier or later induction of ALP in 3D compared to 2D cultures. Moreover, as close inspection of the individual data points potentially indicates two subpopulations with either increased or unchanged ALP activity compared to cells without ES, these results could be explained by donor variability. In contrast to the ALP activity, significantly increased levels of calcium deposition were observed at d21 in cells exposed to ES and osteogenic factors compared to cells in osteogenic medium alone. Interestingly, a significant increase of mineralisation was observed in cells exposed to ES in standard medium compared to unstimulated controls. This might indicate that a basal level of osteogenesis can be induced by ES in 3D, without biochemical osteoinductive cues. In agreement with the results of the Alizarin Red S staining, we detected a significant increase in OCN expression at d21 in cells exposed to ES and osteoinductive cues compared to both cells with ES in standard medium and cells in osteogenic medium without ES. In contrast, a significantly increased expression of OPN has been observed in cells cultivated under osteogenic conditions with ES compared to cells in osteogenic medium alone at d7 and d14, but not at d21. These results are in line with our experiments conducted in 2D, where ES did not change expression levels of OPN at d21. The lack of significant differences at d21 found both in 2D and 3D, could be explained by the fact that OPN is an earlier osteogenic marker than OCN [54].

It is well established that osteogenic differentiation is associated with profound changes in the actin cytoskeleton with increased actin polymerisation and alignment of actin bundles in the periphery [51,52]. A recent study demonstrated that exposure of human BM-MSCs to ES in 2D resulted in actin cytoskeleton rearrangements similar to those induces by osteoinductive factors [28]. Similarly, we demonstrated that ES of ASCs in 3D leads to rearrangement of the actin cytoskeleton towards a denser and more aligned F-actin network independent of the presence of osteoinductive factors. It can be assumed that similar results could be obtained if other adult human stem cell types and biomaterials would be applied in a similar setting. In this context, MSCs from other sources including the bone marrow, umbilical cord, placenta, dental sources, or human neural crest-derived stem cells [47] could be used for developing future therapies for osteoporotic bone fractures [59]. Moreover, in addition to natural hydrogels, blends of electroconductive polymers and collagens can be applied in combination with different sources of human stem cells [60]. However, translating the preclinical results into the clinic will require not only clinically compliant stem cells and biomaterials but also GMP compliant cultivation and differentiation protocols that still need to be developed.

Overall, our in vitro data suggest that 3D cell culture in combination with ES increases osteogenic differentiation potential of ASCs. Future research will investigate if ES of ASCs can induce similar effects in in vivo models of bone regeneration including rodent models and in sheep as a large animal model that represents a closer approximation of bone degeneration and regeneration in humans. Together, data presented in this study and follow-up in vivo data will facilitate development of innovative therapies for osteoporosis and osteoporotic fractures in humans.

## 5. Conclusions

Our study indicates that ES of ASCs in 3D hydrogels composed of aNFC increases osteogenic differentiation in synergy with osteoinductive factors present in osteogenic differentiation medium. Moreover, we provide evidence that ES in 3D aNFC can lead to osteogenic differentiation under standard conditions, although to a lesser extent than in combination with osteogenic factors. Future experiments with optimised parameters for ES and 3D cell culture might lead to a development of new protocols allowing induction of osteogenic differentiation free of exogenous cues and potentially paving the way for improved autologous stem cell-based bone grafts for treatment of osteoporotic fractures.

## Figures and Tables

**Figure 1 biomolecules-10-01696-f001:**
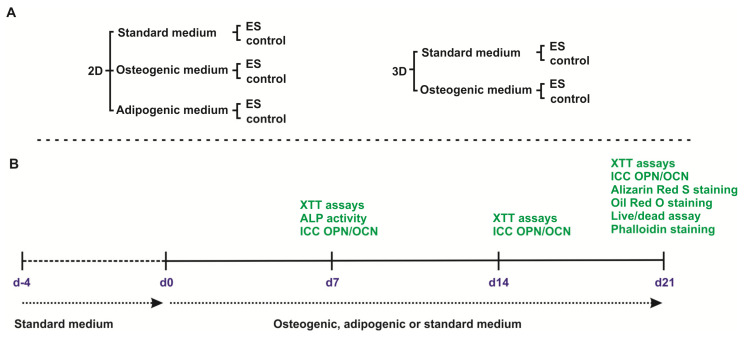
Schematic representation of the experimental design. (**A**) Experimental groups. In 2D, adipose-derived stem cells were differentiated into osteogenic or adipogenic fate or maintained under standard conditions with and without electrical stimulation (ES). In 3D, cells were differentiated into osteogenic fate or kept in standard medium with and without exposure to ES. (**B**) After 4 days of pre-cultivation, medium was replaced by osteogenic, adipogenic, or fresh standard medium. XTT assays and immunocytochemical (ICC) staining against osteopontin (OPN) and osteocalcin (OCN) were performed at day d7, d14, and d21. Alkaline phosphatase (ALP) activity was assessed at d7, whereas Alizarin Red S staining, Oil Red O staining, live/dead assay and phalloidin staining were performed at d21.

**Figure 2 biomolecules-10-01696-f002:**
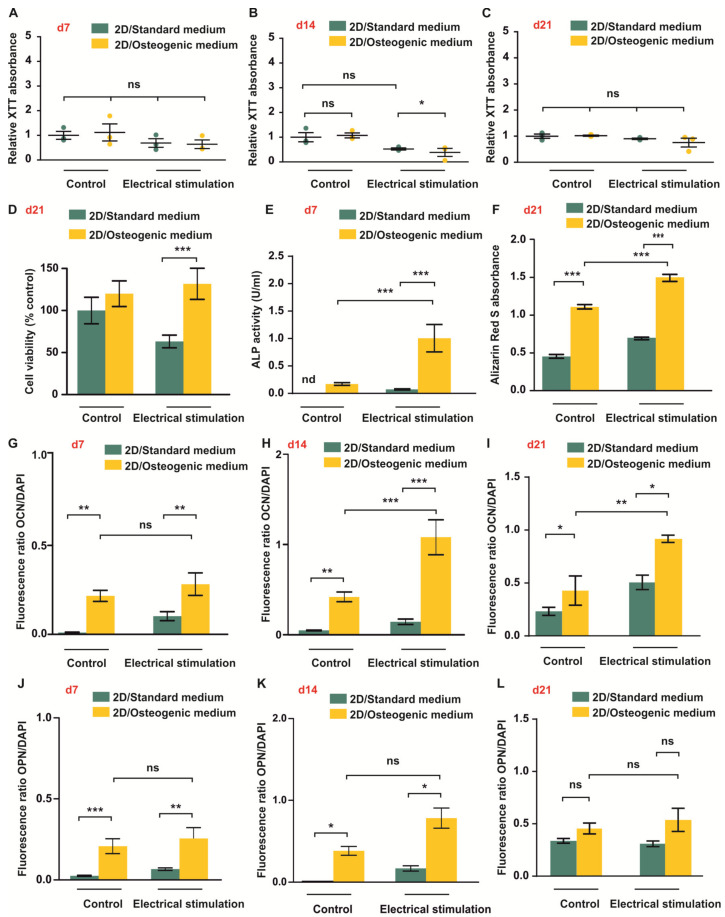
Electrical stimulation (ES) increases osteogenic differentiation of Adipose-derived stem cells (ASCs) under 2D cell culture conditions. (**A**–**C**) ASCs were exposed to ES in standard and osteogenic medium for up to 21 days followed by assessment of viability using XTT assays. At d7 and 21 ES, no significant differences in viability were detected. At d14, ES of cells in osteogenic medium slightly decreased the viability compared to cells exposed to ES in standard medium. (**D**) Calcein/ethidium homodimer-1 based live/dead assay. Cells were cultivated in 2D in standard or osteogenic differentiation medium with or without ES for 21 days followed by a live/dead assay. Data analysis revealed no differences in relative viability in osteogenic differentiation medium compared to control cells and a higher relative viability in cells subjected to osteogenic differentiation and ES compared to cells in standard medium with ES. (**E**) After seven days, cells were assayed for activity of ALP. A significant increase of ALP activity was observed in cells in osteogenic medium exposed to ES compared to non-stimulated cells under osteogenic conditions. (**F**). Mineralisation was assessed by Alizarin Red S assay on d21. ES in osteogenic medium significantly increased the levels of mineralisation compared to cells cultivated under osteogenic conditions without ES. (**G**–**I**) Immunocytochemical analysis of osteocalcin (OCN) at d7, d14, and d21. Cells in 2D were subjected to ES in standard of osteogenic medium for up to 21 days followed by immunocytochemical staining and fluorescence microscopy. At d14 and d21, significantly higher expression levels of OCN were observed in cells in osteogenic differentiation medium exposed to ES compared to osteogenically differentiated cells without ES. (**J**–**L**) Quantification of immunocytochemical analysis of osteopontin (OPN) expression at d7, d14, and d21. No significant differences in OPN expression were found at d21, whereas ES of cells in osteogenic medium significantly increased OPN expression at d7 and d14 compared to cells in osteogenic conditions with and without ES, respectively. No significant differences were found between cells in osteogenic medium with and without ES at all time points. Error bars: SEM. * *p* < 0.05, ** *p* < 0.01, *** *p* < 0.0001. ns: not significant. nd: not detected.

**Figure 3 biomolecules-10-01696-f003:**
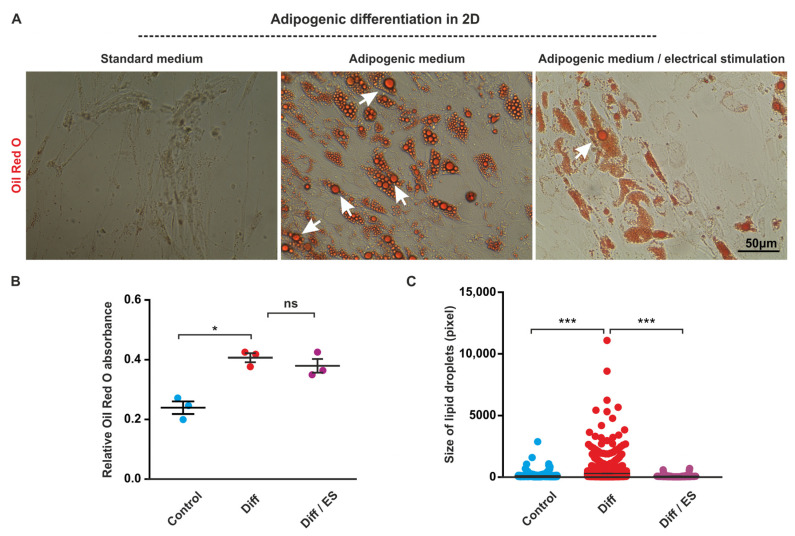
Electrical stimulation (ES) interferes with adipogenic differentiation of adipose-derived stem cells (ASCs). ASCs were cultivated in adipogenic medium or standard medium for 21 days followed by staining of lipid droplets by Oil Red O, microscopy, and image analysis. (**A**) Microscopy revealed multiple large lipid droplets in cells in adipogenic medium without ES and only a few in cells exposed to ES. Scale bar: 50 μm. (**B**,**C**) Image analysis revealed a slight but not significant decrease of Oil Red O absorbance and a highly significant decrease of the average size of the lipid droplets in cells cultivated under adipogenic conditions exposed to ES compared to cells in adipogenic medium without ES. Error bars: SEM. * *p* < 0.05, *** *p* < 0.0001. ns: not significant. Diff: differentiation.

**Figure 4 biomolecules-10-01696-f004:**
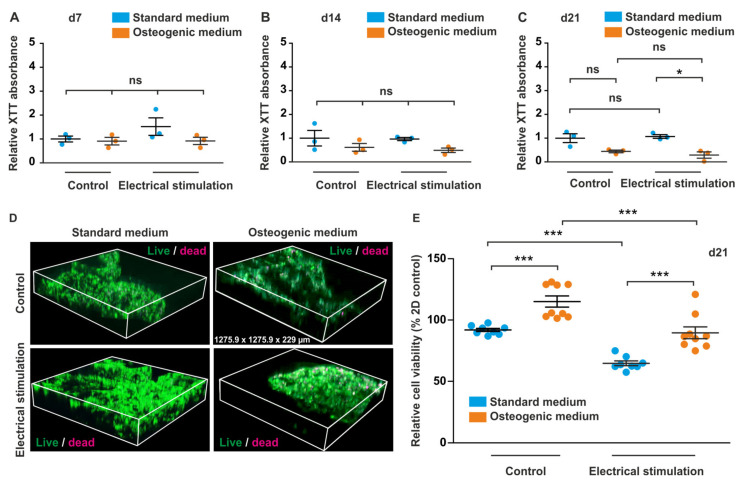
Electrical stimulation (ES) only moderately decreases viability of adipose-derived stem cells (ASCs) in 3D under osteogenic cultivation conditions. (**A**–**C**) Assessment of viability using XTT assays. ASCs embedded into 0.2% aNFC hydrogels were exposed to ES in standard and osteogenic medium for up to 21 days followed by assessment of viability using XTT assays. No significant differences in viability were detected at d7 and d14, whereas a slight but significant decrease was observed in cells exposed to ES in osteogenic medium compared to cells in standard medium with ES. (**D**,**E**) Calcein/ethidium homodimer-1 based live dead assay. Cells were cultivated in standard or osteogenic differentiation medium with or without ES for 21 days followed by a live/dead assay, and confocal laser scanning microscopy with subsequent 3D reconstruction. Data analysis revealed higher relative viability in 3D cells in osteogenic differentiation medium compared to control cells in aNFC and a higher relative viability in cells subjected to osteogenic differentiation and ES compared to cells in 3D with ES. Error bars: SEM. * *p* < 0.05, *** *p* < 0.0001. ns: not significant.

**Figure 5 biomolecules-10-01696-f005:**
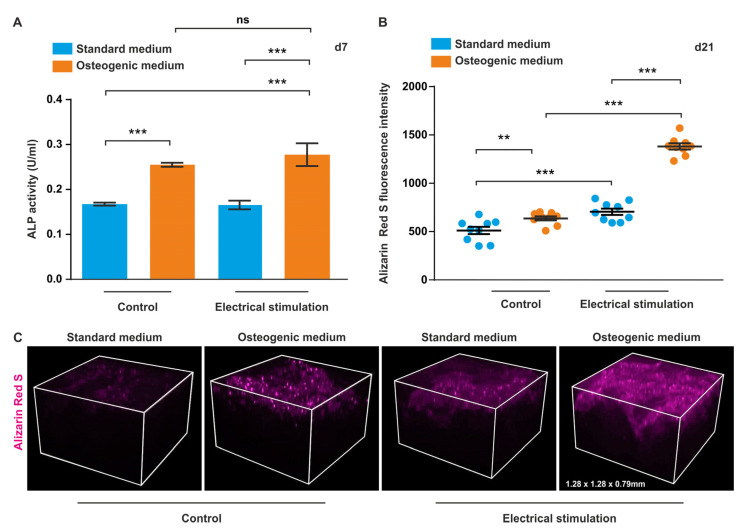
Electrical stimulation (ES) increases alkaline phosphatase (ALP) activity and the level of mineralisation during osteogenic differentiation. (**A**) ALP activity in 3D. ASCs were exposed to ES in osteogenic and standard medium for 7 days followed by an assessment of ALP activity. Significantly higher ALP activities were observed in cells in osteogenic differentiation medium than in standard medium with and without ES. ES resulted in a higher ALP activity in cells in osteogenic medium. (**B**,**C**). Calcium deposition in 3D. ASCs were cultivated in standard or osteogenic cultivation medium with and without ES. After 21 days, cells were subjected to Alizarin Red S staining and confocal laser scanning microscopy. Osteogenic differentiation conditions led to an increase of mineralisation compared to cells in control medium both with and without ES. ES resulted in significantly higher levels of mineralisation in cells cultivated in osteogenic medium. Error bars: SEM. ** *p* < 0.01, *** *p* < 0.0001.

**Figure 6 biomolecules-10-01696-f006:**
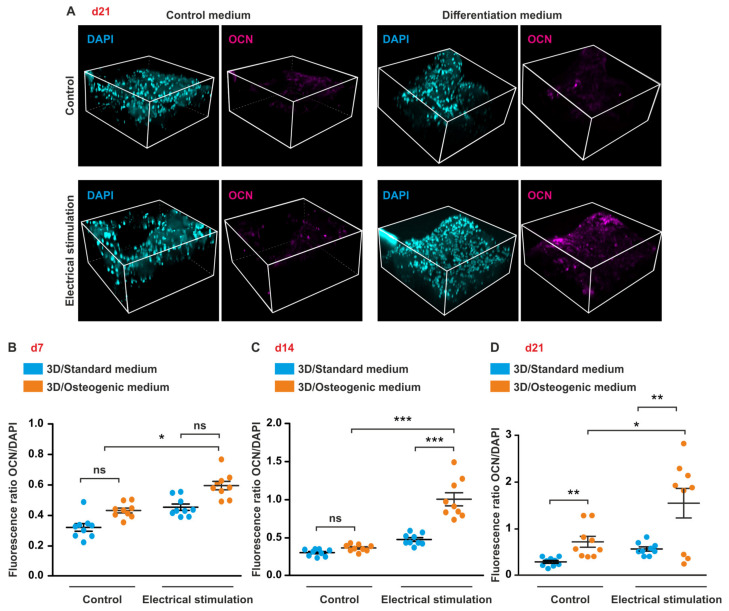
Electrical stimulation (ES) increases expression of osteocalcin (OCN) in 3D. (**A**–**D**) ES increases expression of OCN in 3D. (**A**–**D**) ASCs in osteogenic or standard medium were electrically stimulated for up to 21 days and stained for OCN. Images were taken using confocal microscopy with subsequent 3D reconstruction. ES significantly increased the expression of OCN in cells in osteogenic medium compared to stimulated cells in standard medium and osteogenic medium not exposed to ES at d14 and d21. Error bars: SEM. * *p* < 0.05, ** *p* < 0.01, *** *p* < 0.0001. ns: not significant.

**Figure 7 biomolecules-10-01696-f007:**
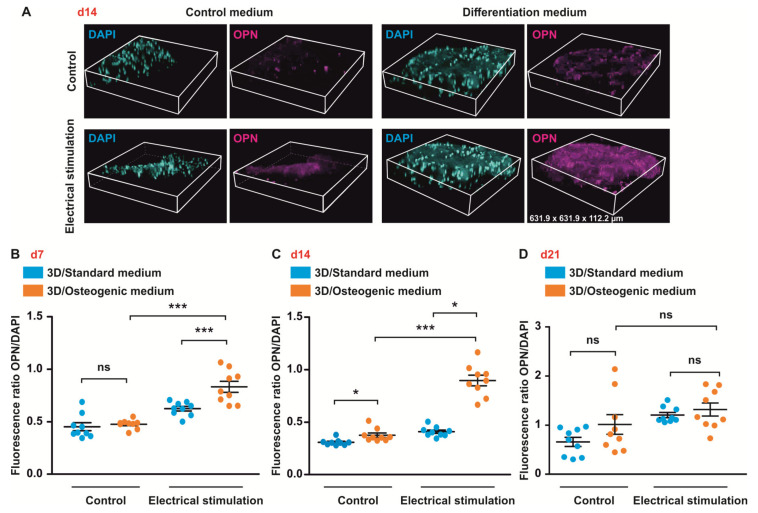
Electrical stimulation (ES) increases expression of osteopontin (OPN) in 3D at d7 and d14. ASCs were cultivated in standard or osteogenic medium with and without electrical stimulation (ES) for up to 21 days. (**A**) 3D reconstruction of z-sections. (**B**–**D**) Quantification of the data. ES resulted in increased expression of OPN in cells under osteogenic conditions at d7 and d14. At d14, higher expression level of OPN was observed in cells in osteogenic medium under ES exposure compared to cells under osteogenic conditions without ES. No significant differences in OPN expression were found at d21. Error bars: SEM. * *p* < 0.05, ** *p* < 0.01, *** *p* < 0.0001. ns: not significant.

**Figure 8 biomolecules-10-01696-f008:**
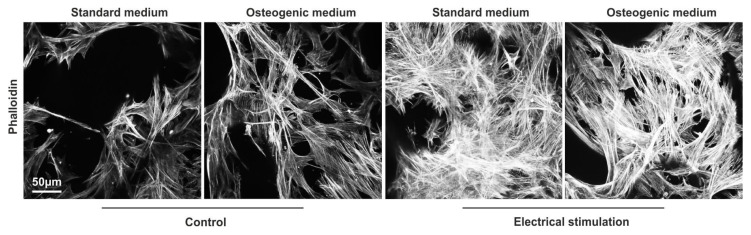
Electrical stimulation (ES) of adipose-derived stem cells (ASCs) induces profound re-arrangements of the actin cytoskeleton. ASCs in anionic nanofibrillar cellulose hydrogels (aNFC) were differentiated in osteogenic differentiation medium for 21 day with and without electrical stimulation (ES) followed by staining of the actin cytoskeleton using phalloidin with subsequent confocal microscopy. Image analysis revealed highly aligned and dense network of F-actin fibres in cells in osteogenic differentiation medium and in cells exposed to ES independent of the culture condition. Images show maximum intensity projections of confocal z-stacks. Scale bar: 50 μm.

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
