# Peer review of "Electrical Stimulation of Adipose-Derived Stem Cells in 3D Nanofibrillar Cellulose Increases Their Osteogenic Potential"

_biomolecules, 2020, doi:10.3390/biom10121696_

Round 1

Reviewer 1 Report

The authors have significantly improved the quality of the manuscript and applied all corrections I inquired. I have no farther critics.

Author Response

We thank the reviewer for their time and their positive evaluation of our manuscript.

Reviewer 2 Report

In the present original work entitled: “ Electrical Stimulation of Adipose-Derived Stem Cells 2 in 3D Nanofibrillar Cellulose Increases their 3 Osteogenic Potential” with the number: biomolecules-1041454, the Authors demonstrated an alternative source for cell-based therapy for the treatment of osteoporosis and osteoporotic features. Researchers used adipose-derived stem cells (ASCs), 3D nanofibrillar cellulose model and electrical stimulation (ES) for up to 21 days. In the background of the evaluation Moreover, Authors relied on the data obtained from 2D model.

At overall, the study seems to be not that innovative but it does demonstrate an interesting solution with widely used human ASCs in clinical scenario. The present study is demonstrated as a pre-reviewed manuscript. Many significant adjustments within the entire MS have been already made and highlighted in red.  

Abstract: There is no reference to any ES conditions (voltage, frequency, time span?) while reading the abstract. Additionally, no information concerning 2D model is included. At least one sentence would be helpful (inviting) for the first impression, before reading the entire Ms.

Intro: very informative part is composed of the information and the expertise of the Authors (ref. 39, 40). From this point of view the design and goal of the study relies on both above mentioned studies using: i) different cell source, ii) 3D biomaterial. At this point the reader expects a strong discussion: why this study is that of high importance to be published in the present Journal. Are there any further downstream experiments planned or extended (in vivo) to allow the study to be close to the pre-clinical scenario?

Here, no short information about 2D model used in the study.

Materials and methods:

Human ASCs: which passage and do the Authors obtained a fully characterization (phenotype) of these cells.

Figure 1. Please comment, why the intervals between the evaluation groups were 7 days. What about the 4 days after the induction onset?

General point: Why these ES conditions… is there any internal reference or from literature..

How the Authors assure that such ES conditions are the best for the present experimental design. How it is possible to move the present data when other biomaterials, cells would be used?

Of note, if this biomaterial (plant hydrogel) has been applied in the clinics (Ref 55), how far the Authors may be able to transfer the present experimental design into other stem cells. Are the media used GLP/GMP-conform? If so, are the data generated suitable to transfer into pre-clinical scenario? This translational task is essential to be answered in context of the suitability of the following 3D model and ES conditions.

To summarize, the entire study is worth to publish in Biomolecules after a minor/major changes.

Author Response

In the present original work entitled: “ Electrical Stimulation of Adipose-Derived Stem Cells 2 in 3D Nanofibrillar Cellulose Increases their 3 Osteogenic Potential” with the number: biomolecules-1041454, the Authors demonstrated an alternative source for cell-based therapy for the treatment of osteoporosis and osteoporotic features. Researchers used adipose-derived stem cells (ASCs), 3D nanofibrillar cellulose model and electrical stimulation (ES) for up to 21 days. In the background of the evaluation Moreover, Authors relied on the data obtained from 2D model.At overall, the study seems to be not that innovative but it does demonstrate an interesting solution with widely used human ASCs in clinical scenario. The present study is demonstrated as a pre-reviewed manuscript. Many significant adjustments within the entire MS have been already made and highlighted in red.  

We thank the reviewer for their time and their valuable suggestions.

Abstract: There is no reference to any ES conditions (voltage, frequency, time span?) while reading the abstract. Additionally, no information concerning 2D model is included. At least one sentence would be helpful (inviting) for the first impression, before reading the entire Ms.

 We have now included this information in the abstract.

To elucidate the osteoinductive potential of a combination of these biophysical cues on ASCs, cells were embedded within anionic nanofibrillar cellulose (aNFC) hydrogels and exposed to electrical stimulation (ES) for up to 21 days. ES was applied to ASCs in 2D and 3D at a voltage of 0.1 V/cm with a duration of 0.04 ms, and a frequency of 10 Hz for 30 min per day.”

Intro: very informative part is composed of the information and the expertise of the Authors (ref. 39, 40). From this point of view the design and goal of the study relies on both above mentioned studies using: i) different cell source, ii) 3D biomaterial. At this point the reader expects a strong discussion: why this study is that of high importance to be published in the present Journal. Are there any further downstream experiments planned or extended (in vivo) to allow the study to be close to the pre-clinical scenario?

We thank the reviewer for this excellent suggestion. We have now incorporated this in the discussion.

Overall, our in vitro data suggest that 3D cell culture in combination with ES increases osteogenic differentiation potential of ASCs. Future research will investigate if ES of ASCs can induce similar effects in in vivo models of bone regeneration including rodent models and in sheep as a large animal model that represents a closer approximation of bone degeneration and regeneration in humans. Together, data presented in this study and follow-up in vivo data will facilitate development of innovative therapies for osteoporosis and osteoporotic fractures in humans.”

Here, no short information about 2D model used in the study.

Information of 2D controls is now provided in the introduction section.

In the present study, we hypothesized that combining 3D cultivation of ASCs in aNFC hydrogels with ES can positively influence their osteogenic differentiation potential. We have shown that ES of ASCs can increase their osteogenic differentiation in 2D whilst decreasing the levels of adipogenic differentiation. This work also identified that ES of ASCs embedded in 3D aNFC hydrogels increases their osteogenic differentiation compared to unstimulated 3D controls. “

Materials and methods:

Human ASCs: which passage and do the Authors obtained a fully characterization (phenotype) of these cells.

This information is now provided.

Figure 1. Please comment, why the intervals between the evaluation groups were 7 days. What about the 4 days after the induction onset?

 We thank the reviewer for this comment. Based on our experience and reports from other groups (Hauser et al, 2012, Greiner et al. 2011, Schürmann et al., 2014, Sun et al., 2007) only low levels of osteogenic differentiation are detectable at d4. Due to the complex experimental design (6 experimental groups in 2D, 4 experimental groups in 3D) we have decided to focus our investigation on the timepoint where significant changes were to be expected. Please note that these timepoint are standard in the field as changes in ALP activity occur at d7, whereas significant changes in expression of OCN, OPN and the levels of mineralisation are to be expected at d14 and d21.

General point: Why these ES conditions… is there any internal reference or from literature. How the Authors assure that such ES conditions are the best for the present experimental design.

We thank the reviewer for this valid question. In 2007, Sun et al assessed the impact of ES with 0.1V-1V/cm on calcium oscillation patterns in MSCs in 2D. They were able to find that 0.1V/cm is sufficient in inducing osteogenic differentiation and also demonstrated that there is no difference between 0.1V and 1V. In our pilot experiments, we have also assessed 0.3V/cm and detected similar increase of osteogenic differentiation. However, this was accompanied by an increase in cell death. Therefore, we have used 0.1V/cm which is sufficient for differentiation without excessive induction of cell death. We have now included a short paragraph in the discussion section and also cite the Sun et al paper.

In 2007, Sun et al exposed human BM-MSCs to ES with 0.1-1V/cm and observed that both conditions result in a decrease of calcium oscillation and an increase of osteogenic differentiation [27]. Importantly, no significant differences in osteogenic induction potential were observed between 0.1V and 1V. Although higher voltages are still used in the field [21,28,29], their long-term application can result in electrochemical reaction in the medium and increased cell death rates [30].”

How it is possible to move the present data when other biomaterials, cells would be used? Of note, if this biomaterial (plant hydrogel) has been applied in the clinics (Ref 55), how far the Authors may be able to transfer the present experimental design into other stem cells. Are the media used GLP/GMP-conform? If so, are the data generated suitable to transfer into pre-clinical scenario? This translational task is essential to be answered in context of the suitability of the following 3D model and ES conditions.

This is now discussed as requested.

“It can be assumed that similar results could be obtained if other adult human stem cell types and biomaterials would be applied in a similar setting. In this context, MSCs from other sources including the bone marrow, umbilical cord, placenta, dental sources, or human neural crest-derived stem cells [47] could be used for developing future therapies for osteoporotic bone fractures [59]. Moreover, in addition to natural hydrogels, blends of electroconductive polymers and collagens can be applied in combination with different sources of human stem cells [60]. However, translating the preclinical results into the clinic will require not only clinically compliant stem cells and biomaterials but also GMP compliant cultivation and differentiation protocols that still need to be developed.”

To summarize, the entire study is worth to publish in Biomolecules after a minor/major changes.

We thank the reviewer again for their time and helpful comments and suggestions.
